# Do Men and Women Have a Different Association between Fear-Avoidance and Pain Intensity in Chronic Pain? An Experience Sampling Method Cohort-Study

**DOI:** 10.3390/jcm11195515

**Published:** 2022-09-20

**Authors:** Sophie Waardenburg, Lars Visseren, Elke van Daal, Brigitte Brouwer, Jan van Zundert, Sander M. J. van Kuijk, Richel Lousberg, Ellen M. M. Jongen, Carsten Leue, Nelleke de Meij

**Affiliations:** 1Department of Anesthesiology and Pain Management, Maastricht University Medical Centre MUMC+, 6229 HX Maastricht, The Netherlands; 2Department of Clinical Epidemiology and Medical Technology Assessment, Maastricht University Medical Centre MUMC+, 6229 HX Maastricht, The Netherlands; 3Faculty of Health, Medicine and Life Sciences, Maastricht University, 6229 ER Maastricht, The Netherlands; 4Department of Psychiatry and Neuropsychology, Maastricht University, 6229 ER Maastricht, The Netherlands; 5Department of Anesthesiology, Multidisciplinary Pain Centre Ziekenhuis Oost Limburg, 3600 Genk, Belgium; 6Faculty of Psychology, Open University, 6419 DJ Heerlen, The Netherlands; 7Department of Psychiatry, Maastricht University Medical Centre MUMC+, 6229 HX Maastricht, The Netherlands

**Keywords:** chronic pain, pain intensity, fear-avoidance, positive affect, negative affect, experience sampling method, momentary assessment, anxiety, depression

## Abstract

Background: Fear-avoidance is one of the factors associated with chronic pain. However, it remains unclear whether the association between fear-avoidance and pain depends on sex. The present study aimed to investigate whether the association between fear-avoidance and pain intensity differed between men and women in chronic pain patients. Additionally, the potential confounding effect of affective experiences on the association between fear-avoidance and pain intensity was analyzed. Method: This cohort study included hospital referred chronic pain patients (*n* = 45). Short momentary assessment questions according to the experience sampling method (ESM) were used to repeatedly assess patients’ pain intensity, level of fear-avoidance and positive as well as negative affect during their daily life. Linear mixed-effects models were applied in the statistical analysis. Unadjusted and adjusted models were made, in which the latter corrected for statistically significant affective experiences and baseline variables, taking the Aikake Information Criterion into account to assess a better model of fit. Results: The results demonstrated an association between fear-avoidance and pain intensity that differed for men and women. In men (*n* = 13), no association between these variables was found (−0.04 (95% CI: −0.14, 0.06) with a *p*-value of 0.48), whereas in women (*n* = 32), an increase in fear-avoidance was associated with a (slight) increase in pain intensity (0.18 (95% CI 0.06, 0.30) with a *p*-value of 0.003). Affect did not confound the above-mentioned findings. Conclusion: Our data supports previous research highlighting the importance of sex differences in pain experience. These findings may be relevant for clinicians to consider more personalized (i.e., gender specific) pain management in chronic pain patients.

## 1. Introduction

Chronic pain affects more than 30% of people worldwide and has a large impact on both patients and society [1]. Due to the complex interactions between biological, psychological and social factors [1,2], it is difficult to manage chronic pain. One of these factors is fear-avoidance, which refers to the avoidance of movements or activities resulting from fear of pain [3]. According to the fear-avoidance model, pain may be interpreted as threatening (i.e., pain catastrophizing), which can lead to avoidant behaviors and hypervigilance to bodily sensations. Conversely, fear-avoidance and hypervigilance may induce physical disuse and disability, contributing to long-term consequences, including maintenance of chronic pain disability or an increase in the pain experience [3]. Although the association between fear-avoidance and chronic pain has been well-established, only sparse research has been conducted on potential sex differences regarding this association. As a growing number of articles suggests the importance of sex differences in relation to pain, and specifically in pain catastrophizing [4,5,6], it is important to further investigate whether the association between fear-avoidance and pain also depends on sex.

Furthermore, the biopsychosocial model of pain shows that emotional distress or affective states may influence pain intensity [1] and may, therefore, also confound the association between sex and fear-avoidance. It is known that dynamic fluctuations regarding positive and negative affect are observed in various mental disorders [7], such as depression. Given that depression and pain share pathways [8], fluctuations in emotion regulation as observed in depression and other mental disorders may also be found in chronic pain patients. However, the effect of affective states, such as happiness, anxiety and irritation on the association between fear-avoidance and pain has not been adequately studied. The cross-sectional design of studies that have investigated the association between affective states and chronic pain could not capture the fluctuations of emotional distress over time.

Hence, the present study aimed to investigate whether the association between fear-avoidance and pain intensity in chronic pain patients differs between men and women. Additionally, the potential confounding effect of specific affective experiences on the association between fear-avoidance and pain intensity was analyzed by using the experience sampling method (ESM).

## 2. Methods

### 2.1. Study Design

This cohort study used questionnaires administered according to ESM. Experience sampling is a structured digital diary technique to appraise subjective experiences in daily life, often applied in patients with psychiatric disorders or somatic illnesses [9]. Patients are repeatedly asked to complete short questionnaires during the day, which allows for the assessment of moment-to-moment changes in both symptoms and mental states, aiming to map daily functioning [10]. This study was approved by the local medical ethical committee (METC-number: 2018-0955).

### 2.2. Study Population

The cohort of the present study consisted of chronic pain patients who were referred to the University Hospital Pain Centre of the Maastricht University Medical Centre+ (MUMC+). The patients were recruited from March 2019 until July 2021 while performing their digital intake at the pain center, during which they were asked whether they wanted to be approached for participation in this study. If their answer was positive, patients were contacted by a research nurse for a more extensive explanation about the ESM-procedures. Patients with any type of pain at any location were eligible for participation. To be included, patients had to be 18 years or older and to have experienced pain complaints for at least three months. Additionally, the patient had to be in possession of a smartphone and able to use the ESM application named Psymate. Patients who were interested in participation also received all required information by an information letter, complemented with a consent form. Before the start of the study, all patients who wanted to participate provided informed consent.

### 2.3. Experience Sampling

Both outcome (pain intensity), predictor (fear-avoidance) and potential confounders (affect) were measured by repeated ESM assessments. These ESM assessments consisted of 18 questions and were completed through a smartphone application (Psymate). The items in the Psymate application illustrate adequate psychometric properties, and sensitivity to change over time [11]. Patients were asked to answer the questions 10 times a day, for six consecutive days. The questionnaires were completed in semi-random time blocks of 112.5 min from 7:30 a.m. until 10:30 p.m. during the patients’ daily life, whenever patients received a notification (‘beep’) from the Psymate-app on their smartphone [12,13]. Fear-avoidance was assessed by the statement ‘due to fear for (more) pain I did not make unnecessary movements since the last beep’, asking the participants about their fear-avoidance behavior since the last beep. The items of positive and negative affect come originally from the validated PANAS questionnaire [14,15,16] and were assessed thoroughly before the application in the ESM. Positive affect was assessed by the following statements ‘I feel cheerful’, ‘I feel relaxed’, ‘I feel satisfied’, and ‘I feel enthusiastic’, whereas negative affect was measured by the statements ‘I feel insecure’, ‘I feel irritated’, ‘I feel lonely’, ‘I feel anxious’, ‘I feel guilty’ and ‘I am worrying’. The 10 different items concerning the affective state, as well as the item assessing the level of fear-avoidance, were answered on a 7-point Likert scale, ranging from 1 (not at all) to 7 (very much). The outcome variable ‘pain intensity’ was assessed by the statement ‘I am in pain’, and could be answered on an 11-point scale, ranging from 0 (no pain) to 10 (worst pain possible).

### 2.4. Baseline Variables

As part of the standard digital intake at the MUMC+, patients were asked to complete a set of questionnaires that reflected the pain complaints, quality of life, anxiety and depressive symptoms. These questionnaires consisted of the Hospital Anxiety and Depression Scale (HADS), Numeric Rating Scale (NRS) for pain intensity, Pain Catastrophizing Scale (PCS), Brief Pain Inventory (BPI) and the 12-item Short-Form Health Survey (SF-12) [17,18,19,20,21]. An explanation of how these measurement instruments were assessed is provided in more detail by the article of Brouwer et al. [22]. During intake, patients also had to indicate how long they had been experiencing pain and at which location(s). Moreover, demographic variables including sex, age, marital status, education level and employment were collected. In addition to the intake questionnaires, patients had to complete one additional questionnaire that assessed the level of fear-avoidance at baseline. The ‘TAMPA Scale for Kinesiophobia’ (TSK) (Dutch translated), which includes 17 questions on a 4-point scale, was used for this. TSK-scores range from 17 to 68, and scores greater than 37 indicate a high degree of fear-avoidance [23]. Similar to the ESM-measurements, the TSK was completed through the Psymate-app once before the start of the ESM-examination period.

### 2.5. Statistical Analysis

Baseline characteristics of the cohort are described as mean and standard deviation for continuous variables, and as count and percentage for categorical variables. Sex differences in baseline characteristics were tested using the independent-samples t-test for continuous variables, and Pearson’s chi-square test or Fisher’s exact test for categorical variables. ESM-data were analyzed using linear mixed-effects models with random intercept and slope on three levels; patients, days, and beeps. The model was built in several steps. First, the crude association between fear-avoidance and pain was assessed as fixed and as random effect. Second, the interaction of fear avoidance and sex was added. The third and fourth model assessed for potential confounders concerning baseline variables and affect. Consequently, two backward stepwise elimination processes were applied. The third model assessed the first backward stepwise elimination of the baseline variables (patients sociodemographic variables, pain characteristics and PROMs of Table 1). The fourth model assessed the items of negative and positive affect (‘I feel cheerful, ‘I feel relaxed’, ‘I feel satisfied’, ‘I feel enthusiastic’, ‘I feel insecure’, ‘I feel irritated’, ‘I feel lonely’, ‘I feel anxious’, ‘I feel guilty’ and ‘I am worrying’) as being potential confounders by the backward stepwise elimination process. Autocorrelation by using a first-order continuous time covariate autoregressive structure was added in the fifth model. The stipulated models are presented in Figure 1. Analyses were performed using R, version 4.1.2, with the function lme (linear mixed effects models) from the statistical package nlme (3.1–153). All tests were investigated two-sided against a significance level (α) of 0.05.

## 3. Results

### 3.1. Description of the Sample

Initially, 217 patients indicated they were interested in the study and were therefore approached. Out of these 217 patients, 168 patients (77%) declined to participate after receiving all the information about the study procedures, whereas 49 patients (23%) provided informed consent. Three patients were excluded from analysis because their pain complaints were present for less than three months, and one patient was excluded due to missing data on sex at baseline. This resulted in a sample of 45 chronic pain patients, from which 13 (21%) were men and 32 (71%) women (Figure 2). The mean level of fear-avoidance (TSK) at baseline was significantly (*p* = 0.013) higher for men (39.6; SD ± 6.5) than for women (34.8; SD ± 5.2). Moreover, a high degree of fear-avoidance (TSK-score > 37) was also more frequently present in men (69%) than in women (38%), although not significantly different (*p* = 0.053). Mean pain intensity (NRS) was 6.8 (SD ± 1.9) for men and 7.3 (SD ± 1.6) for women (*p* = 0.391), indicating no statistically significant sex difference in pain intensity at baseline. Other baseline variables, as well as the *p*-values of the differences between men and women, are presented in Table 1.

### 3.2. Sex Differences in the Association between Fear-Avoidance and Pain Intensity and the Influence of Affective States

The crude association between fear-avoidance and pain intensity had a coefficient of 0.17 (95% CI: 0.12, 0.22), *p* = 0.000, indicating that an increase in fear-avoidance of 1 unit was associated with an average pain increase of 0.17. The model that also included the interaction between fear-avoidance and sex showed that the association differed between men and women: the interaction term had a coefficient of 0.18 (95% CI: 0.05, 0.31), *p* = 0.005 (Table 2; model 2). For men, a 1-point increase in fear-avoidance was associated with a −0.02 decrease in pain intensity, whereas for women a 1-point increase in fear-avoidance was associated with a 0.18 increase in pain intensity (Table 2 and Figure 3).

In the subsequent model, potential confounders were added. Backward stepwise elimination resulted in a model with the baseline variables ‘age’ and ‘lower leg’ and affective experiences ‘relaxed’, ‘irritated’, and ‘satisfied’ included (Table 2; model 5, and Figure 4). By adding the confounders, the association between fear-avoidance and pain intensity in the model with the interaction did not change considerably from a coefficient of −0.02 (95% CI: −0.12, 0.09) with a *p*-value of 0.78 to a coefficient of −0.04 (95% CI:−0.14, 0.05) with a *p*-value of 0.48. Although these three affective experiences all had a significant association with pain intensity, the estimate of the interaction term between fear-avoidance and sex did not change by adding affect to the model (Table 2; model 5).

## 4. Discussion

### 4.1. Summary of Main Findings

To our knowledge, this is the first study using the experience sampling method to investigate sex differences in the association between fear-avoidance and pain intensity in chronic pain patients, including the potential confounding effect of affective experiences. Cross-sectional results demonstrated that men had on average more fear-avoidance than women. However, results from the longitudinal data of the ESM suggest that no association between fear-avoidance and pain intensity was found in men, whereas in women, an increase in fear-avoidance was associated with a (slight) increase in pain intensity. Nonetheless, affect did not confound these findings.

### 4.2. Differences in the Association of Fear-Avoidance and Pain Intensity

The fact that men had a higher mean TSK-score than women in the present study is consistent with the literature from previous cross-sectional studies that investigated sex differences in TSK-scores concerning chronic pain patients [24,25]. It remains debatable why male chronic pain patients tend to have more fear-avoidance than female patients, although it has been suggested that this could depend upon social norms, higher expectations or a deeper concern about losing work capacity or productivity as a result of re-injury [25]. However, the results of our study indicate that the tendency of having more fear-avoidance does not seem to influence pain intensity in men. Moreover, whether the increase in fear-avoidance in men at baseline influences (negatively) pain treatment outcomes remains unanswered.

### 4.3. Sex Differences in the Association between Fear-Avoidance and Pain Intensity

Both the unadjusted and adjusted model concerning the interaction between fear-avoidance and sex in relation to pain intensity showed that this interaction was significant, and hence, the association between fear avoidance and pain differs between men and women. The adjusted model was corrected for the affective experiences ‘relaxed’, ‘irritated’, and ‘satisfied’, but did not lead to a different conclusion. In the unadjusted and adjusted models, the association between fear-avoidance and pain intensity for men was negligible (0.02 and −0.04, respectively). In contrast, for women, the model demonstrated that the association between fear-avoidance and pain intensity was equal to a coefficient of 0.18 in both models (Table 2), indicating that increases in fear-avoidance were associated with (slight) increases in pain intensity. Whether this (small) association was clinically significant, it may yet be debated. We propose to further investigate if this association holds when applied to other pain populations, preferably with larger sample sizes and equal percentage of both sexes.

Ramirez et al. in 2014 [26] analyzed differences in pain experience between men and women in patients with spinal chronic pain and found a contrasting result, in that fear-avoidance was associated with pain intensity in men, but not in women. However, because of the cross-sectional design of the study the strength of the evidence is limited. Moreover, previous studies suggest that women are more sensitive to threat-related stimuli than men, and this would generally lead to an increased pain perception [6,27] and have greater catastrophic thoughts than men, which would generally lead to an increased pain perception. The results found in the present study are in line with these suggestions.

No previous studies have investigated the potential confounding effect of affective states on the association between fear-avoidance and pain intensity with the ESM. In a review by Baets et al. in 2019 the predictive moderating and mediating roles of emotional factors were examined on pain and disability following shoulder treatment [28]. A predictive role was found for fear-avoidance of pain and disability when surgical treatment was given, yet not when receiving physiotherapy. Moreover, this study indicated a moderating role for optimism in the relationship between catastrophizing and shoulder disability in patients receiving physiotherapy. However, this role was not found in the relationship between fear-avoidance and disability of the shoulder. The results of our ESM study specified that affect has a moderating effect on pain intensity itself, but not on the relationship between fear-avoidance and pain intensity. The statistically significant effect of positive affective experiences, such as feeling relaxed (−0.15, *p* ≤ 0.001) and satisfied (−0.10, *p* ≤ 0.001), on pain intensity itself may indicate that there is a potential role for positive affect, such as optimism, self-efficacy and positive expectations in future research and treatment [28,29].

### 4.4. Strengths and Limitations

The present study has a few important advantages. First, due to the use of the ESM, symptoms were assessed in the actual moment, eliminating the potential influence of recall and contextual biases, which is a common problem with traditional retrospective questionnaires [30,31]. Moreover, symptoms such as pain and fear, as well as affect, are likely to fluctuate over time [7]. Due to the many repeated measurements in ESM, these fluctuations could be captured, in contrast to cross-sectional studies. Because of these advantages and the low cost of the ESM method, it might be an attractive and effective method to use more often in future (clinical) studies, or even treatment trajectories, since ESM is feasible due to the widespread use of smartphones. Moreover, ESM may be applied as an additional tool in clinical practice to provide feedback as part of personalized pain intervention [32].

On the other hand, this study has a few limitations. First, seventy-two percent of the participants completed the full 6 days from the ESM examination-period, which resulted in 28% missing data. As experience sampling is time-consuming, these missed assessments were expected beforehand, and the repeating character of ESM accounts for, and decreases, the influence of missing data [33]. However, missed assessments might be a concern, as a sub-group of pain patients might have missed assessments as a consequence of their current mood or level of pain. This may have resulted in overestimation of functioning [9]. Moreover, the sample size in this study was rather small, with an especially low number of men. The percentage of 29% of men deviates from the 40% of men in the overall pain registry cohort DATA*PAIN* [22]. Accordingly, a lack of power could explain why no significant association was found between fear-avoidance and pain intensity for men. Many patients who initially indicated to be interested in the study chose not to participate after receiving all information about the study procedures (Figure 1). This indicates that ESM may be (too) burdensome, at least with the current number of questions and repeated measures. As the usability of ESM in chronic pain patients has not yet been validated, it remains difficult to conclude whether this method is suitable for the chronic pain population. Although momentary assessment is recommended in different somatic and psychiatric conditions, and the benefits of the ESM are becoming more and more apparent [34], it is important to perform more research about ESM and to evaluate its validity and reliability in chronic pain patients.

Fear-avoidance was assessed by the statement ‘due to fear for (more) pain I did not make unnecessary movements since the last beep’, asking the participant how the behavior of fear has influenced the level of movement since the last beep. As a result, a time frame is assessed between the afore-appointed beep until the actual beep, representing a lagged item. This was the main reason why we did not add a lagged model, as in our case that would be regressing two time points in time instead of one. Moreover, as mentioned before, no intention of causality was intended, meaning that the direction of predictor and outcome could have been reversed: an analysis we want to recommend for future research.

Furthermore, even though the dataset covered a vast number of relevant factors for chronic pain, some factors such as pain etiology were not accounted for at baseline, and other important factors such as pain catastrophizing were missing in the daily assessments, which could explain the sex differences found in our results.

## 5. Conclusions

The results in this study indicate that the association between fear-avoidance and pain intensity differs between men and women. For men, no association between these variables was found, whereas for women, an increase in fear-avoidance was associated with a (slight) increase in pain intensity. Affective experiences, however, did not confound the association between fear-avoidance and pain intensity in either men or women. Our findings support research highlighting the importance of sex differences in pain experience, which may be important for clinicians to consider for a more personalized pain management approach in chronic pain patients. Nevertheless, further research with a larger sample and equal numbers of sexes is needed to confirm these findings and their clinical implication.

## Figures and Tables

**Figure 1 jcm-11-05515-f001:**
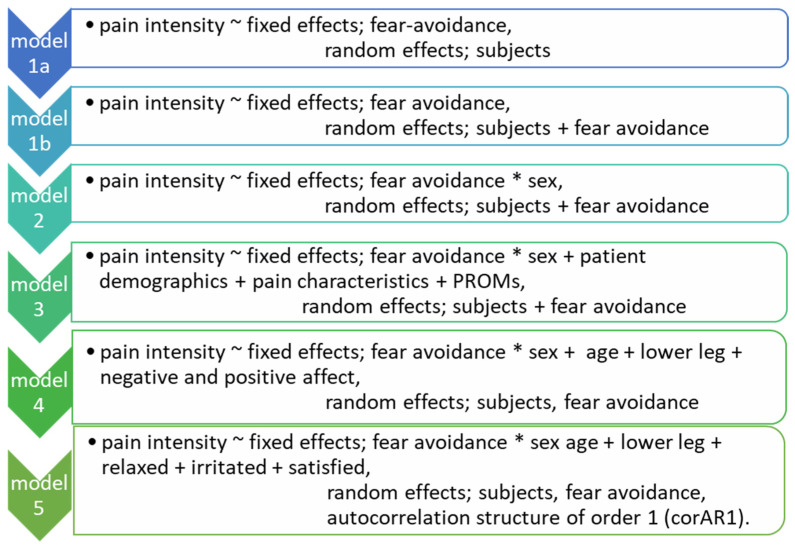
Construction of linear mixed-effects model applied to the data. ~ Separation of the dependent and independent variables. * Indicative of an interaction term and the original variables themselves.

**Figure 2 jcm-11-05515-f002:**
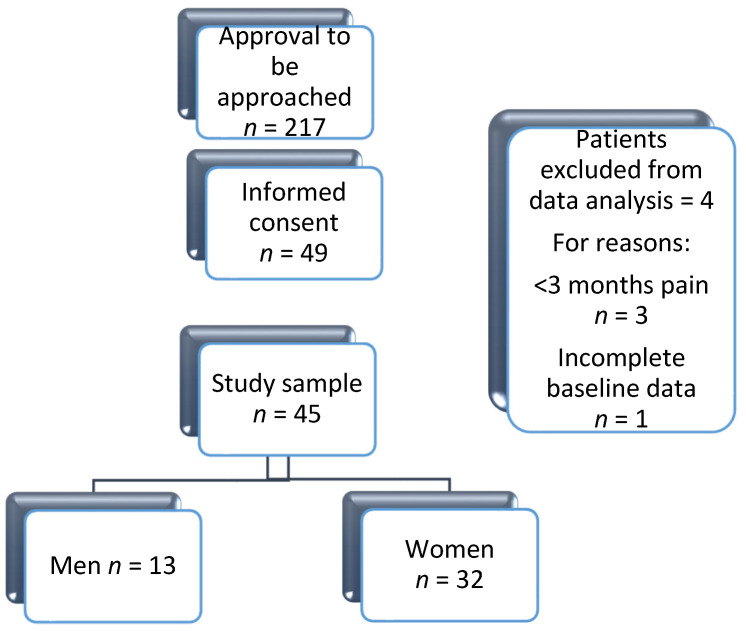
Flowchart of the study sample.

**Figure 3 jcm-11-05515-f003:**
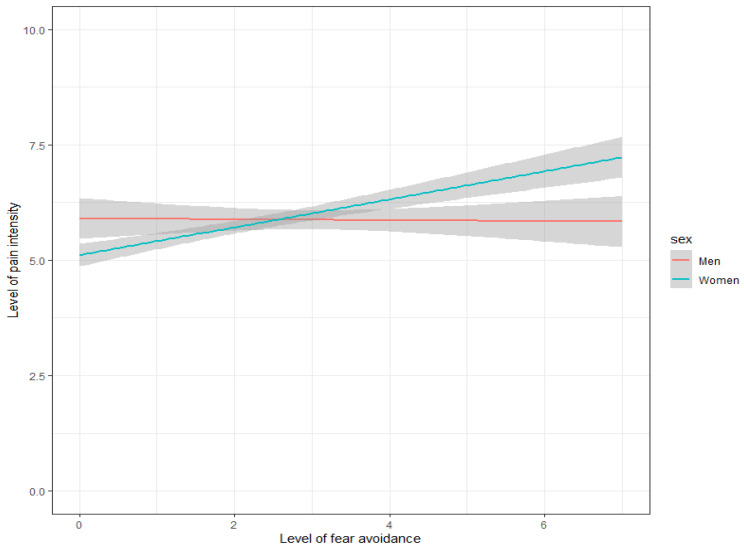
Visualization of the interaction between fear-avoidance and sex in relation to pain intensity. Note: the grey area is the 95% confidence interval of the estimates.

**Figure 4 jcm-11-05515-f004:**
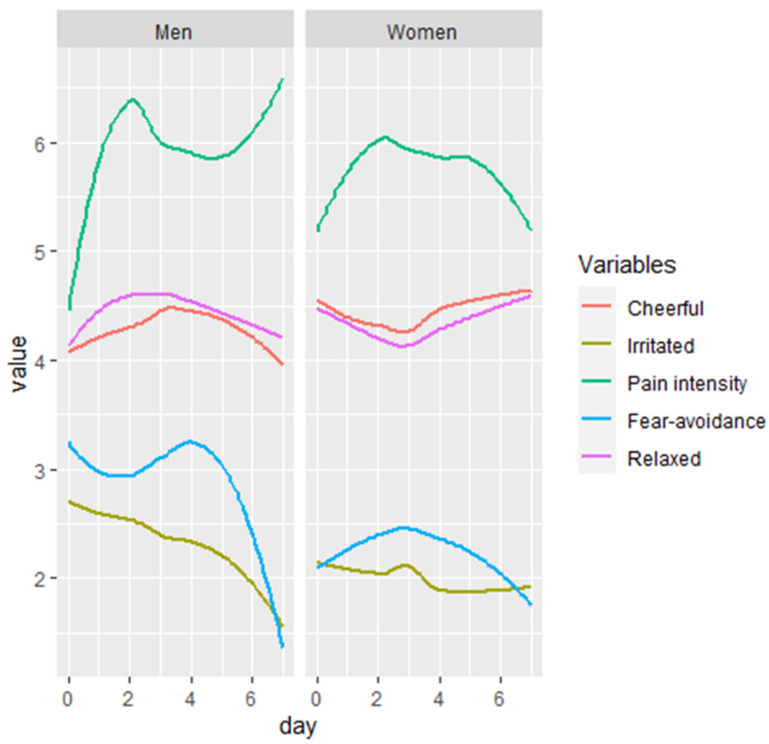
Visualization of the fluctuation of pain intensity, fear avoidance and affective experiences for men and women separately. Note: fear-avoidance and the affective experiences are measured on a 7-point Likert scale and pain intensity is measure on the eleven-point numeric rating scale. Each line represents the average of the 10 beeps per day per variable.

**Table 1 jcm-11-05515-t001:** Baseline description of the chronic pain patient cohort.

Patient Baseline Characteristics	Total Cohort, *n* = 45	Men, *n* = 13	Women, *n* = 32	*p*-Value
*Demographic Characteristics*				
Age in years, *mean (SD)*	47.6 (12.8)	52.8 (13.8)	45.5 (12.0)	0.086
Marital status, *n (%)*				0.411
Relationship	36 (80.0)	9 (69.2)	27 (84.4)	
No relationship	9 (20.0)	4 (30.8)	5 (15.6)	
Education, *n (%)*				0.287
Low *(<9 years of education)*	32 (71.1)	11 (84.6)	21 (65.6)	
High *(≥9 years of education)*	13 (28.9)	2 (15.4)	11 (34.4)	
Employment, *n (%)*				1.000
Unemployed *(no paid job)*	29 (64.4)	8 (61.5)	21 (65.6)	
Employed *(paid job)*	16 (35.6)	5 (38.5)	11 (34.4)	
*Pain Characteristics*				
Pain duration in months, *mean (SD)*	73.2 (81.1)	45.9 (55.5)	84.3 (87.8)	0.088
Pain location, *n (%)*				
Head	5 (11.1)	1 (7.7)	4 (12.5)	1.000
Neck	15 (33.3)	3 (23.1)	12 (37.5)	0.492
Arm	7 (15.6)	1 (7.7)	6 (18.8)	0.654
Lower back	25 (55.6)	10 (76.9)	15 (46.9)	0.066
Upper leg	19 (42.2)	5 (38.5)	14 (43.8)	0.745
Lower leg	12 (26.7)	3 (23.1)	9 (28.1)	1.000
Chest/abdomen	4 (8.9)	2 (15.4)	2 (6.3)	0.567
Other	10 (22.2)	2 (15.4)	8 (25.0)	0.698
*PROMs Scores*				
NRS, *mean (SD)*	7.1 (1.7)	6.8 (1.9)	7.3 (1.6)	0.391
PCS, *mean (SD)*	23.2 (11.9)	26.8 (14.1)	21.8 (10.9)	0.207
BPI REM *mean (SD)*	11.6 (8.1)	14.2 (9.6)	10.5 (7.2)	0.095
BPI WAW, *mean (SD)*	24.7 (10.0)	25.6 (9.2)	24.3 (10.5)	0.440
TSK, *mean (SD)*	36.2 (6.0)	39.6 (6.5)	34.8 (5.2)	0.013 *
TSK > 37, *n (%)*	21 (46.7)	9 (69.2)	12 (37.5)	0.053
HADS-A, *mean (SD)*	6.8 (3.8)	8.2 (4.2)	6.2 (3.6)	0.125
HADS-D, *mean (SD)*	7.5 (4.9)	9.2 (5.2)	6.8 (4.7)	0.127
PHS, *mean (SD)*	29.4 (6.8)	29.6 (6.4)	29.2 (7.0)	0.862
MHS, *mean (SD)*	45.7 (12.1)	43.1 (12.5)	46.8 (12.0)	0.367

Abbreviations: NRS, Numerical Rating Scale for pain intensity; PCS, Pain Catastrophizing Scale; BPI-REM, Affective Subscale of the Brief Pain Inventory; BPI-WAW, Active Subscale of the Brief Pain Inventory; TSK, Tampa Scale of Kinesiophobia; HADS-A, Hospital Anxiety and Depression Scale-Anxiety subscale; HADS-D, Hospital Anxiety and Depression Scale-Depression subscale; PHS, Physical Health Score; MHS, Mental Health Score; PROM, Patient Reported Outcome Measure. * *p*-value < 0.05.

**Table 2 jcm-11-05515-t002:** Unadjusted and adjusted model regarding sex differences in the association between fear-avoidance and pain intensity.

	Model 2 AIC = 4476.42			Model 5 ^a^ AIC = 4376.42		
	Estimate	CI	Sig.	Estimate	CI	Sig.
Intercept	9.08	6.47, 11.7	0.000 ***	9.52	6.86, 12,18	<0.001 ***
Fear-avoidance	−0.02	−0.12, 0.09	0.78	−0.04	−0.14, 0.06	0.48
Sex (men = 0; women = 1)	−0.53	−1.96, 0.89	0.45	−0.4	−1.82, 1.02	0.57
Fear-avoidance x sex	0.18	0.05,0.31	0.005 **	0.18	0.06, 0.30	0.003 **

Dependent variable: pain intensity; CI = confidence intervals; ^a^ Adjusted for baseline variables: age, lower leg and the emotions: relaxed ***, irritated *** and satisfied ***; ** *p*-value < 0.01; *** *p*-value < 0.001.

## Data Availability

All data relevant to the study are included in the article. Data can only be obtained by contacting the corresponding author.

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
