# Peer review of "Do Men and Women Have a Different Association between Fear-Avoidance and Pain Intensity in Chronic Pain? An Experience Sampling Method Cohort-Study"

_jcm, 2022, doi:10.3390/jcm11195515_

Round 1
Reviewer 1 Report
Overall this is a novel, interesting, well-performed and analyzed study. The findings are weak, though significant and suggestive and the sample size is small. All of these limitations are noted by the authors.
What is necessary to address is to provide more information regarding the LMM analysis and the validity of the items used in the ESM. What statistical package in R was used (e.g. nlme, lme, glmm, etc)?
How was model selection performed? Aside from study design characteristics, were simpler models tested to determine if they sufficiently explained the data before the authors used an autoregressive model? What factors were assigned to the random effects portion and what factors to the fixed effects portion of the model? What are the authors justification for factor assignment?
How does the difference in TSK between the sexes effect the overall findings? The author's should include this as a fixed effect as it is a potential confound.
What validated questionnaires, if any, are the items in the ESM derived? Is there any evidence of psychometric validity to support using these items alone of grouped together? If these items are from standardized and previously validated questionnaires please provide citations for the Dutch validity studies.
There are several minor typos, English usage and style errors, especially regarding punctuation around citations and style of citations. I would estimate about two or three errors per paragraph, although these errors are generally minor and the text reads well overall. I will not enumerate these errors at the moment, but the authors are advised to improve their proof reading.
In the results and discussion more neutral language should be used to described the associations discovered in the study between pain intensity and fear avoidance. It must not be forgotten or glossed over that causal influence is in no way implied in these results (e.g. "having more fear-avoidance does not seem to influence pain intensity in men."). The presumed direction of causality from fear-avoidance to pain seems implied throughout.
Author contributions section and funding are not completed.
Author Response
We would like to thank you for the time and effort you have put into this revision and giving us the possibility to revise and improve our manuscript. In the attached document the full revision can be found.
With kind regards,
Sophie Waardenburg

Reviewer 2 Report
Thank you for the opportunity to review this manuscript describing sex differences in the association between fear avoidance and pain intensity in chronic pain, using ESM data. Findings describe a small association between daily fluctuations in fear-avoidance and pain intensity in woman but not men, even after controlling for some affective states. Authors conclude that a sex difference exist in this fear-avoidance-pain relationship. Here are my comments.
- Daily fluctuations in pain intensity and the relationships between pain and movement highly depend on pain etiology (neuropathic vs nociceptive), thus, with such a small sample size of a mixed chronic pain population and uneven group distribution, the sex differences found could simply be a difference in pain etiology between men and women in the sample. If this information is available, then the authors need to account for pain etiology in their analysis. If not available, this should be discussed as an important limitation.
- There seem to be a discrepancy between the methods and results in terms of adjusting for confounding factors. Methods state: “Backward stepwise elimination was applied to analyze which baseline variables (patients sociodemographic variables and pain characteristics) and those representing affective distress had to be included in the model.” However, results state “ In the subsequent model, the potential confounding affective states were added. Backward stepwise elimination resulted in a model with the affective experiences…” Were baseline measures accounted for? None were significant? This needs better reporting.
- Pain catastrophizing is an integrative part of the fear-avoidance model and expected to influence the pain-fear of movement relationship; arguably more so than negative affect. Thus, the role of pain catastrophizing should be included in the background/discussion of the manuscript. As per my previous comment, was baseline PCS considered in the model and was found not significant? This is not clear in the results.
-Also, pain catastrophizing differs between sex and also mediate sex differences in pain, so is sex differences found in this study due to differences in day-to-day catastrophizing? Unfortunately catastrophizing was not part of the ESM but this possible explanation of the findings should at least be discussed.
- Authors should describe/justify their choice of ESM measure for fear-avoidance which seems to reflect more activity avoidance (movement = pain) than somatic focus (pain = re-injury). Reporting correlations with TSK baseline scores (and subscales) would re-assure readers about its validity. Especially since participants scored high on the TSK but the ESM measured fairly low fear-avoidance.
- Fluctuations in pain intensity have been shown to follow a circadian rhythm for some individuals with some evidence in animal models that this may be different between sex. Thus, days may be more suitable as a fixed factor. In fact, authors collected 10 beeps per day for only 6 days suggesting that within day was more an interest than day-to-day fluctuations.
- Authors do not report a priori sample size calculation, nor power calculation. More attention to the STROBE reporting guidelines should be taken.
- The clinical importance of the findings need to be discussed. A 2 point change in pain intensity is considered clinically important, thus, a 0.18 change per 1 unit of fear-avoidance (out of 6), in women with low fear-avoidance (according to the ESM), have little clinical importance.
Specific comments
- Abstract: Add Ns per sex.
- Figure 2. Add which error estimate was used for the shaded area.
- Figure 3. Add more details regarding line graph representation (e.g. Are each line representing daily avg scores or all 10 beeps).
Author Response

(The authors gave the same response as above.)

Round 2
Reviewer 2 Report
The authors addressed with satisfaction most of my comments, in large part with the added clarification of the analyses performed. My only outstanding comment is related to the integration/acknowledgment of pain catastrophizing in the manuscript. Here are my suggestions:
In intro:
- Add “ (i.e. pain catastrophizing)” after “… pain may be interpreted as threatening…”.
- Acknowledging sex difference exists specifically in pain catastrophizing near the end of first paragraph (citing Fillingim 2009 review) would strengthened the justification to investigate sex differences in fear-avoidance since these are theoretically related.
In discussion,
- Edit the following statement: “Moreover, previous studies suggest that women are more sensitive to threat-related stimuli (Ramírez-Maestre, Martínez et al. 2004, Rhudy and Williams 2005) and have greater catastrophic thoughts (Sullivan, 2001) than men, which would generally lead to an increased pain perception.”
- For the added limitation statement: “Furthermore, even though the dataset covered a vast amount of relevant factors for chronic pain, some factors such as pain etiology was not accounted for at baseline, or other important factors such as pain catastrophizing were missing in the daily assessments.’ Add “… which could explain the sex differences found in our results.” at the end of the statement.
Other minor revisions
- The added statement “The items in the Psymate application illustrate adequate psychometric properties, and sensitivity to change over time (Verhagen, Berben et al. 2017).” should come either before or after the description of all ESM items, instead of between the description of fear-avoidance and affective state items, as this statement is relevant to all items.
- Consider a table for the added description of different models.
- Reiterate in the conclusion that further research with a larger sample and equal Ns of sexes is needed to confirm these findings and their clinical implication.
Author Response
Dear reviewer,
Thank you very much for your time and interest in reviewing this manuscript and the opportunity to improve our manuscript. We agree on the suggestions that you have provided and are therefore included in the manuscript. Hence only the new version of the manuscript is submitted.
With kind regards,
Sophie Waardenburg
